# In Vitro Evaluation of the Cytotoxic Effect of *Streptococcus pyogenes* Strains, Protegrin PG-1, Cathelicidin LL-37, Nerve Growth Factor and Chemotherapy on the C6 Glioma Cell Line

**DOI:** 10.3390/molecules27020569

**Published:** 2022-01-17

**Authors:** Alexandr N. Chernov, Anna Tsapieva, Diana A. Alaverdian, Tatiana A. Filatenkova, Elvira S. Galimova, Mariia Suvorova, Olga V. Shamova, Alexander N. Suvorov

**Affiliations:** 1Scientific and Educational Center “Molecular Bases of Interaction of Microorganisms and Human”, Center for Personalized Medicine, FSBSI Institute of Experimental Medicine, Acad. Pavlov Street, 12, 197376 St. Petersburg, Russia; lero269@gmail.com (T.A.F.); elya-4@yandex.ru (E.S.G.); oshamova@yandex.ru (O.V.S.); alexander_suvorov1@hotmail.com (A.N.S.); 2Department of Medical Biotechnologies, University of Siena, 53100 Siena, Italy; ditanka17@gmail.com; 3Sechenov Institute of Evolutionary Physiology and Biochemistry of the Russian Academy of Sciences, pr. Thorez, 44, 194223 St. Petersburg, Russia; 4Disease Systems Immunology, Department of Biotechnology and Biomedicine, Technical University of Denmark, Søltofts Plads 224, DK-2800 Kgs. Lyngby, Denmark; marsuv@dtu.dk

**Keywords:** glioma C6, human normal fibroblasts, cytotoxicity, *Streptococcus pyogenes*, NGF, LL-37, PG-1, chemotherapy, cathelicidin family peptides, MTT assay, trypan blue assay, real-time xCELLigence system

## Abstract

Brain cancer treatment, where glioblastoma represents up to 50% of all CNS malignancies, is one of the most challenging calls for neurooncologists. The major driver of this study was a search for new approaches for the treatment of glioblastoma. We tested live *S. pyogenes*, cathelicidin family peptides and NGF, assessing the oncolytic activity of these compounds as monotherapy or in combination with chemotherapeutics. For cytotoxicity evaluation, we used the MTT assay, trypan blue assay and the xCELLigence system. To evaluate the safety of the studied therapeutic approaches, we performed experiments on normal human fibroblasts. Streptococci and peptides demonstrated high antitumor efficiency against glioma C6 cells in all assays applied, surpassing the effect of chemotherapeutics (doxorubicin, carboplatin, cisplatin, etoposide). A real-time cytotoxicity analysis showed that the cell viability index dropped to 21% 2–5 h after *S. pyogenes* strain exposure. It was shown that LL-37, PG-1 and NGF also exhibited strong antitumor effects on C6 glioma cells when applied at less than 10^−4^ M. Synergistic effects for combinations of PG-1 with carboplatin and LL-37 with etoposide were shown. Combinations of *S. pyogenes* strain #7 with NGF or LL-37 demonstrated a cytotoxic effect (56.7% and 57.3%, accordingly) on C6 glioma cells after 3 h of exposure.

## 1. Introduction

Oncological diseases are an urgent medical and social problem. According to the World Health Organization (WHO) forecasts, by the year 2040, the number of oncology cases in the world will increase to 29.5 million people per year compared to 18.1 million people in 2018 [1,2]. Malignant brain neoplasms in adults include glioblastomas and astrocytomas [3]. The prognosis for these patients is unfavorable.

Therapy for central nervous system tumors is commonly carried out using chemotherapy with agents such as temozolomide, cisplatin, carboplatin and bevacizumab [4,5,6]. The mechanism of action is based on the alkylation of the DNA in the case of cisplatin and on methylation in temozolomide [7,8]. However, the use of chemotherapy is accompanied by several problems: (1) poor permeability of the blood–brain barrier for chemotherapy; (2) high cytotoxicity, causing adverse side reactions in the physiological system, often leading to the termination of chemotherapy; (3) the non-selective action of chemotherapy resulting in the apoptosis of healthy cells; (4) the multiple drug resistance of tumor and stem cells of malignant tumors, which often leads to metastases and tumor recurrence in treated patients [9,10,11].

In this regard, the issue of the development of new drugs and the search for new pharmacological substances with antitumor activity against tumors with primary and acquired drug resistance are urgent [12]. Endogenous regulatory proteins—growth factors and cationic antimicrobial peptides (AMPs) of the human innate immunity system or their combination with chemotherapy drugs as well as live bacteria or bacterial bioactive substances—can be considered as such potential candidates in oncology.

To date, more than eighty growth factors have been described [13]. The earliest and most well-studied of these is NGF which is known to have the ability to increase the survival of the neural cells and also to inhibit both angiogenesis and tumor invasion [14,15]. The perspective of the antitumor effect of NGF and the combination of NGF with chemotherapy on glioma cell cultures has been studied insufficiently.

More than 5000 AMPs are known to date [16]. Most AMPs are molecules consisting of 12–50 amino acids with a high content of arginine and/or lysine, possessing antimicrobial activity against bacteria, unicellular fungi, protozoa and viruses. AMPs exhibit immunomodulatory, mitogenic, antitumor and less toxic effects on normal tissues [16,17,18,19].

The presence and severity of peptides’ effects depend on the structural features of each peptide. To study the mechanisms of antitumor action, we selected two peptides with different structures from the cathelicidin family: LL-37, with an α-helical structure from azurophilic granules of human neurophils, and protegrin-1 (PG-1), a peptide with a β-hairpin conformation from porcine neutrophils. LL-37 is localized predominantly in the secretory granules of neutrophils and is released upon their activation. In high concentrations, these peptides are toxic to human cells [16,20], which creates an obstacle for their introduction into medicine. One of the ways to solve this problem may be the reduction of the effective concentration of these substances, which can be reached by considering the joint use of AMP with chemotherapy drugs to identify their synergistic effects. Such antitumor efficacy of the combinations remains unexplored in relation to tumor cells. The study of the features of the combined effect of AMP or growth factors with chemotherapy drugs can help to reveal the most promising combinations for practical use.

Another experimental approach for tumor treatment is the usage of live bacteria with oncolytic properties. It has been shown that some *Salmonella* spp., *Enterococcus* spp., *E. coli* and several other species as well as some bacterial substances have great anti-tumor potential [21,22]. Among them, *Streptococcus pyogenes* (group A streptococci or GAS) are promising agents due to oncolytic effects shown by “Coley’s toxin” [23] and OK-432 [24]. The cytotoxic activity of the *Streptococcus pyogenes* strain GUR on murine malignant tumor cells was shown earlier [25,26]. In order to develop new strategies for the tumor treatment, we studied the effects of NGF, PG-1, LL-37, their combinations with chemotherapy and live oncolytic bacteria on the C6 glioma cell culture.

## 2. Results

### 2.1. Evaluation of the Cytotoxic Effect of Streptococcus pyogenes on C6 Glioma Cells

The cytotoxic effect of GAS on C6 glioma cells was studied using the MTT assay, the trypan blue exclusion test of cell viability and the xCELLigence system for real-time cytotoxicity analysis (Table 1).

The data of Table 1 show that *S. pyogenes* strain #21 has the highest cytotoxic activity against C6 glioma cells in the MTT test and trypan blue assay (*p* < 0.05). The differences between the data obtained in the MTT test and in the trypan blue assay can be explained by their registration at different times of exposure. The MTT test data show the cytotoxic efficacy of streptococci 2.5 h after addition and in the trypan blue test after 4 h. These values are confirmed by the time curve obtained experimentally using the xCELLigence system (Figure 1).

The data of Figure 1 show that all three studied streptococcal strains demonstrated fast oncolytic action against the C6 cell line and confirmed previous data obtained in the MTT and trypan blue assays (Table 1). *S. pyogenes* 21 inhibited the growth of C6 glioma cells by 78.7% (compared to the control) as early as 2 h after addition and its effect remained constant during the rest of the observation time (8 h). *S. pyogenes* 7 showed a slower cytotoxic effect than *S. pyogenes* 21 and the cell inhibition rate reached 79.3% after 3 h of exposure. *S. pyogenes* GUR and *S. pyogenes* GURSA1 demonstrated the slowest cytotoxic effects on C6 glioma cells along with the highest cell inhibition rates of 81.8% and 79.3% after 4 and 6 h, respectively. Moreover, the cytotoxic effect of the *S. pyogenes* GUR and GURSA1 strains slightly increased over time reaching growth inhibition rates of 85.0% and 81.5%, respectively, after 8 h.

### 2.2. Evaluation of the Cytotoxic Effect of Protegrin-1, Cathelicidin LL-37 and NGF on C6 Glioma Cells

In order to determine the cytotoxic antitumor efficacy of NGF, PG-1, and LL-37 on C6 glioma cells, their effect was compared by IC50 and by the efficacy of chemotherapy drugs according to the results of the MTT assay and trypan blue staining (Table 2).

As indicated in Table 2, NGF (IC50 0.0148 and 0.0025 μM) showed the greatest antitumor efficacy in C6 glioma cell cultures. LL-37 and PG-1 also had a strong antitumor effect, superior to the effect of chemotherapy drugs, both according to the results of the MTT assay (1.07 and 10.1 μM) and as a result of the trypan blue staining (1.14 and 8.6 μM). The difference in absolute values is explained by different mechanisms for assessing cytotoxicity underlying these two methods; when using the MTT test, the level of cell metabolism is measured, whereas the trypan blue assay measures exactly the number of living and dead cells.

According to the guidelines for preclinical drug testing, a compound of a new class is considered cytotoxic at IC_50_ ≤ 10^−4^ M, and an analogue of a known anticancer drug is considered cytotoxic if it is IC_50_ ≤ IC_50_ of the reference drug [27]. Since the IC_50_ of NGF, LL-37, and PG-1 are significantly less than 10^−4^ M, they can therefore be considered cytotoxic active compounds with antitumor activity. The cytotoxic effect of NGF, PG-1 and LL-37 on C6 glioma cells observed in real time using the xCELLigence system is shown in Figure 2.

The results presented in Figure 2 demonstrate that LL-37, PG-1, NGF and TMZ have a rather strong oncolytic effect with CIs of 28.1%, 30.4%, 42.1% and 29.9%, respectively.

### 2.3. Determination of Chemotherapy Action on the C6 Cell Line

As a control study, the cytotoxic action of standard chemotherapy drugs towards C6 cells over time is shown in Figure 3.

As expected, all the chemotherapy drugs demonstrated a strong oncolytic effect: cisplatin and doxorubicin with a CI of 39%, etoposide with a CI of 84.6%, carboplatin with a CI of 74%, and temozolomide with a CI of 32%. On the other hand, cells in control wells continued to grow after the addition of DMEM (negative control). However, the effect of the drugs was much slower than the GAS action or LL-37, PG-1, and NGF. Only temozolomide’s oncolytic action was as urgent as the studied substances.

### 2.4. Determination of the Combined Action of S. pyogenes 7 and S. pyogenes 21 with PG-1, LL-37, NGF and Temozolomide on the C6 Cell Line

We selected *S. pyogenes* 7 and *S. pyogenes* 21 strains as having the fastest cytotoxic effect on C6 glioma and studied their effect in combination with PG-1, LL-37, NGF and TMZ in real time when added at the stationary phase of cell line growth (Figure 4).

The results presented in Figure 4 show that the combination of streptococci and peptides did not show any significant synergetic cytotoxic effect on the C6 cell line. Moreover, the combination of *S.pyogenes* 7 with all the studied substances as well as the combination of *S.pyogenes* 21 with NGF and TMZ showed a complete absence of any oncolytic effect. The combination of *S. pyogenes* 21 with PG-1 and LL-37 suppressed tumor growth by 65% and 68%, respectively, whereas *S. pyogenes* 21 by itself reached a 79.5% death rate of C6 cells during real-time observation (Figure 1) and caused the death of 79.2% of C6 cells in the MTT and trypan blue assays (Table 1).

### 2.5. Evaluation of the Cytotoxic Combined Effect of NGF, Protegrin-1 and Cathelicidin LL-37 with Chemotherapy on C6 Glioma Cells

To establish the cytotoxic antitumor efficacy of NGF, PG-1, and LL-37 combinations with chemotherapeutic agents on C6 glioma cell culture, we compared their IC50 with the IC50 of chemotherapy using an MTT assay and trypan blue staining (Table 3 and Table 4).

MTT assay data in Table 3 show that the IC_50_ values of LL-37 combinations with doxorubicin, carboplatin, cisplatin, and etoposide were lower (1.3, 4.4, 20.3 and 18.5 times, respectively) than the IC50 of the chemotherapy drugs alone, which indicates a stronger cytotoxic antitumor effect of these combinations. At the same time, only the IC_50_ of the combination of LL-37 with etoposide was 1.7 times lower than that of cathelicidin. This combination had the highest cytotoxic efficacy compared to other combinations against C6 glioma cells.

The IC_50_ values of the combinations of LL-37 with temozolomide and PG-1 and NGF with all tested chemotherapy drugs were higher than the IC_50_ values of the chemotherapy drugs alone. This indicates a lower cytotoxic efficiency of the combinations of PG-1 and NGF with selected chemotherapy drugs and LL-37 with temozolomide in the culture of C6 glioma cells. Additionally, the IC_50_s of combinations of PG-1 and NGF with all tested chemotherapy drugs, as well as LL-37 with doxorubicin, carboplatin, temozolomide, and cisplatin were higher than the IC_50_s of PG-1, LL-37, and NGF.

The results of the trypan blue tests in Table 4 state that only the IC_50_s of the combinations of PG-1 with doxorubicin (3.27 times) and cisplatin (2.91 times) were lower than the IC_50_ of these chemotherapy drugs in comparison with the rat C6 glioma culture. The IC_50_ values of the combinations of LL-37 and NGF with all chemotherapy drugs and PG-1 with temozolomide were higher than the IC_50_s of the tested chemotherapy drugs, PG-1, LL-37 and NGF, which indicates a weaker cytotoxic antitumor effect of the combinations compared to the action of the chemotherapy drugs, NGF, and peptides alone. The IC50 of the combination of PG-1 with carboplatin did not differ from the IC50 of the chemotherapy drug.

The CI values and action of the combinations of PG-1, LL-37, and NGF with chemotherapy on the C6 glioma according to the results of the MTT assay are presented in Table 5.

The CI values (<1) in Table 5 show that synergism was observed only for the combinations PG-1 + carboplatin and LL-37 + etoposide on C6 glioma cells. Additivity (CI = 1) was observed for the combination of PG-1 + temozolomide. All other combinations of substances showed antagonism (CI > 1) in comparison with the use of chemotherapy drugs and peptides alone. For the combination PG-1 + carboplatin, the dose reduction index (DRI) was 2.93, and for LL-37 + etoposide, the DRI was 119.8.

The CI values and action of the combinations of PG-1, LL-37, and NGF with chemotherapy on the C6 glioma according to the results of the trypan blue staining are presented in Table 6.

Table 6 shows that, in line with trypan blue assay results, only combinations of PG-1 with doxorubicin and cisplatin exhibited synergistic cytotoxic effects (CIs of 0.65 and 0.60, respectively) compared to the chemotherapy drugs’ effects on the C6 glioma cells. The DRI was also calculated for the combinations PG-1 + doxorubicin and PG-1 + cisplatin, which were 6.12 and 3.20, respectively. Notably, all combinations of chemotherapy drugs with LL-37 and NGF showed very strong antagonism (CI > 10) in the culture of C6 glioma.

Furthermore, on the C6 culture, the combined effects of PG-1 and LL-37 peptides with each other and with NGF were assessed in comparison with the IC of peptides and NGF alone at doses close to the IC50 revealed in the MTT and trypan blue assays (Figure 5 and Figure 6).

The results in Figure 5 show that the effectiveness of the combinations PG-1 + LL-37, PG-1 + NGF and PG-1 + LL-37 + NGF was lower (*p* < 0.05) compared to the IC of PG-1. The effects of the combinations PG-1 + LL-37, LL-37 + NGF and PG-1 + LL-37 + NGF did not differ significantly from the IC of LL-37. However, the effectiveness of PG-1 + NGF and PG-1 + LL-37 + NGF were higher (*p* < 0.05) than the IC of NGF.

The data in Figure 6 indicate that the effectiveness of the combinations of PG-1 + LL-37, PG-1 + NGF, and PG-1 + LL-37 + NGF was lower (*p* < 0.05) compared to the IC of PG-1. The effects of the combinations PG-1 + LL-37 and PG-1 + LL-37 + NGF were significantly (*p* < 0.05) lower than the IC of LL-37. The effectiveness of the combinations LL-37 + NGF and PG-1 + LL-37 + NGF was also lower (*p* < 0.05) than the IC of NGF. The IC of PG-1 + NGF did not differ significantly from the IC of NGF and PG-1.

### 2.6. Evaluation of the Cytotoxic Effect of NGF, Protegrin-1, Cathelicidin LL-37 and Streptococcal Strains on Normal Fibroblasts Cells

In order to evaluate the possible cytotoxic effect of the studied substances against normal cells, we studied the action of the peptides and streptococci towards normal human fibroblasts. The results of the study are presented on Figure 7 and Figure 8.

The data in Figure 7 indicate that PG-1, LL-37, and NGF have no statistically significant cytotoxic effect on the cell culture of normal human fibroblasts.

The data in Figure 8 indicate that the GAS strains *S. pyogenes* 7, *S. pyogenes* 21, *S. pyogenes* GUR, and GURSA1 do not have a statistically significant effect on the survival of cells of normal human fibroblasts.

## 3. Discussion

The cytotoxic antitumor effect of the live streptococci *S. pyogenes* 7, *S. pyogenes* 21, *S. pyogenes* GUR and *S. pyogenes* GURSA1 as well as the cytotoxic effect of NGF, LL-37, and PG-1 on the rat glioma C6 cell culture was demonstrated using the MTT assay, the trypan blue exclusion test of cell viability and the xCELLigence system for real-time cytotoxicity analysis. All three assays applied showed similar results, confirming the reliability of the obtained data.

The studied GAS strains demonstrated a strong ability to suppress C6 cell line growth causing the death of 78.7–85% of the cells after 8 h exposure. *S. pyogenes* 21 inhibited the growth of C6 glioma cells by 78.7% already 2 h after administration and was the most vital strain in this experiment. *S. pyogenes* 7 showed a slower oncolytic effect than *S. pyogenes* 21 and the cell inhibition rate reached 79.3% after 3 h of exposure. *S. pyogenes* GUR and *S. pyogenes* GURSA1 demonstrated the slowest oncolytic effect on C6 glioma cells but the final cell death rate after 8 h of the exposure with these microorganisms reached 85.0% and 81.5%, respectively.

The data described in the current manuscript correspond to the different studies on the oncolytic effects of streptococci [28,29,30,31,32]. German scientists from the Rostock University Medical Center showed that the arginine deiminase (35 mU/mL) in *Streptococcus pyogenes* 591 (serotype M49) induces cell death (HROG02, HROG05) by autophagy as a result of the activation of genes encoding heat shock proteins. In addition, *S. pyogenes*’ arginine deiminase enhanced the effects of curcumin, resveratrol, quinacrine, and sorafenib when administered in combination with them for 72 h. [28]. Yoshida J. et al. found that *S. pyogenes* acid glycoprotein (SAGP) induces DNA fragmentation and the apoptosis of fibrosarcoma A cells through the GTP-binding protein, protein tyrosine phosphatase, and the inhibition of the EGFR/p42,44 MAPK cascade [29,30]. Hyaluronidase in bacteriophage H4489A (HylP) *S. pyogenes* and its mutant HylPH262A cleaves the hyaluronan of the extracellular matrix and inhibits the invasion and proliferation of breast cancer cells (Hs578T, MDA-MB-231 and MCF-7) [30]. *S. pyogenes* Su can induce the formation of oxygen radicals and hydrogen peroxide in Ehrlich carcinoma cells [32]. Although the studied GAS strains demonstrated promising oncolytic properties, further studies should be carried out to clarify the mechanisms of the oncolytic properties of *S. pyogenes* strains 7, 21, GUR and GURSA1 and to discover the pathways and the substances responsible for the cytotoxic properties of these strains against tumor cells.

The results obtained for the combined action of streptococcal strains with peptides against C6 cells revealed the absence of oncolytic action in most of the combinations. Only *S. pyogenes* 21 with NGF and *S. pyogenes* 21 with LL-37 demonstrated some cytotoxic effect but at least 20% less than GAS or peptides alone. This can be explained by the antimicrobial nature of the peptides and their ability to cause the damage of microbial cells as well.

As for NGF, the results we obtained in C6 glioma cells partially agree with data of Watanabe et al. [33], according to which exogenous NGF (100 ng/mL, 3.37 × 10^−9^ M) decreased the number of C6 cells by day 4 by 53.4 ± 5.15% compared with the control, and induced signs of morphological differentiation. The latter discrepancy can be explained by the different nature and form of the protein used [33]. As shown by the results of Singer H. et al., NGF acts as an antimitogenic protein only in the case where C6 glioma stably expresses the TrkA receptor [34].

The differences between the IC50 values in the MTT assay and the test with trypan blue for the same combinations can reflect the different degree of involvement of cellular processes with mitochondrial dehydrogenases as well as the damage to the cell membrane, as assessed in these tests. In the case where the IC50 of the combination in the MTT assay was lower than that for staining with trypan blue, such as PG-1 or LL-37 with doxorubicin, it indicates the role of mitochondrial dehydrogenases in the realization of the cytotoxic effects of these combinations of substances. On the contrary, when the IC50 of the combination in the MTT test was higher than the IC_50_ in the test with trypan blue, for example, PG-1 with cisplatin or etoposide, this indicates the predominant participation of cell plasmalemma damage in the cytotoxic action of the combination.

C6 glioma cells are more resistant to doxorubicin (IC_50_ = 18.5 μM) than, for example, to etoposide (IC_50_ = 6.93 μM). In this case, the degree of chemoresistance of C6 cells correlated with a decrease in the accumulation of doxorubicin in glioma cells expressing P-glycoprotein (ABCB1 protein) [35]. We obtained similar data which demonstrate that C6 glioma is more sensitive to etoposide (16.6–19.7 times according to MTT and trypan blue assays) compared to doxorubicin.

Our data indicate a strong antitumor cytotoxic effectiveness of LL-37 and PG-1 on C6 glioma cells (Table 3). LL-37 binds to transmembrane domains of purinergic metabotropic P2 × 7 and four types of G-protein-coupled receptors (GPCRs): N-formylpeptide-2 receptor (FPR2), CXC chemokine-2 (CXCR2), Mas-associated gene X (MrgX2), purinergic (P2Y11) receptors that activate p44/42MAPK, and phosphatase-1 of mitogen-activated protein kinase (MKP1) [36]. Tumor cells undergo apoptosis due to the disruption of the integrity of their membranes by peptides, likely inducing the activation of apoptotic factors through the G-protein-coupled receptor (GPCR): AIF, Bax, Bak, Puma, and p53, as shown in HCT116 colorectal cancer cells [37]. Colle et al. studied the dose-dependent one-day effect of LL-37 (0–50 μM) and its LL17–32 fragment on the viability of human GB U87G cells [38]. Using the MTT assay, we determined the IC_50_ dose of cathelicidin LL-37 to be 12.5 μM.

Coelho Neto et al. studied the expression of 14,000 genes influenced by LL-37 and inhibited by small interfering RNAs in the SKBR3 cells of breast cancer and A375 human melanoma [39]. Only the expression levels of 14 genes (*NAT1, TERT, PARD6A, UTF1, HSPA9, FGF2, HAND1, SOX15, HNF4A, TCF3, GDF3, NODAL, CDC42, OLIG2, FOXA2*) differed by more than 1.5 times in comparison with the control [39]. Most of these genes are expressed in cancer stem cells and are associated with the development of tumor chemoresistance. It is possible that the expression patterns of these genes can also influence the sensitivity of glioma cells to LL-37.

Our results indicate a strong antitumor effect of PG-1 on cell cultures of C6 glioma cells (Table 3). In other types of tumor cells, it was shown that PG-1 induces the formation of transmembrane pores, causing an intracellular flow of Ca^2+^, which, in turn, triggers the activation of the protein and gene p53, caspase-3, an inhibitor of cyclin-dependent kinase 1A (CDKN1A) and the inhibition of the proliferating cell nuclear antigen (PCNA), leading eventually to apoptosis [40]. The regulation of these genes inhibits the passage of the cell cycle from G1 to S phase, which blocks the division of cancer cells and tumor growth [41]. PG-1 can interact with the transmembrane domains of MrgX2 and receptors for insulin-like (IGF1R) and epidermal growth factors (EGFR) [42,43], as well as induce the expression of p53 protein and apoptotic genes such as CDKN1A (inhibitor of cyclin-dependent kinase 1A) or p21 [41].

## 4. Materials and Methods

### 4.1. Streptococcus pyogenes Strains

For the study, four strains of GAS with previously discovered oncolytic properties were chosen: the *S. pyogenes* strains GUR, GURSA1, 7, and 21. *S. pyogenes* GUR is a throat isolate from a scarlet fever patient, which was kindly provided by prof. Chereshnev V.A., Perm State University. *S. pyogenes* GURSA1 is a derivative *S. pyogenes* GUR strain with an inactivated M-protein encoding gene [25]. *S. pyogenes* strains GUR and GURSA1 previously showed cytotoxic activity towards murine malignant tumor cells [25,26]. *S. pyogenes* strains 7 and 21 were selected during preliminary studies on the cytotoxic activity of several GAS strains towards C6, U-251 and A-549 cell lines. The strains were cultivated in Todd–Hewitt broth (Condalab, Madrid, Spain) for 16 h at 37 °C from single colonies. The optical density of overnight cultures was measured at 600 nm to estimate the number of bacterial cells using previously made calibration curves. Cultures were centrifuged at 6000× *g* for 5 min, washed with PBS once and resuspended in DMEM to obtain an equal concentration of bacterial cells. For all the tests, 10^6^ CU/mL *S. pyogenes* in DMEM was used.

### 4.2. Cell Culture

The cell culture of rat C6 gliomas was obtained from the Russian Collection of Cell Cultures, Institute of Cytology, Russian Academy of Sciences (Saint-Petersburg, Russia). C6 rat glioma cells were cultured at 2.5 × 10^5^/mL in Petri dishes (d = 35 mm, Nunc, Allerod, Denmark) or 96-well flat-bottomed plates (TPP, Geneva, Switzerland, 1 × 10^4^/well) in Dulbecco’s Modified Eagle Medium (DMEM, Sigma-Aldrich, Darmstadt, Germany) containing 10% fetal bovine serum (Sigma-Aldrich) and 10^−4^ g/mL gentamicin sulfate (Shandong Weifang Pharmaceutical Factory Co., Liaocheng, China) in a Heracell CO_2_ incubator (Thermo Fisher, Saint-Petersburg, Russia) at 37 °C with 95% humidity and 5% CO_2_ for 1–2 days [44]. Fibroblast cell culture was obtained from the cell bank of Pokrovskaya Hospital (Saint-Petersburg, Russia). Human fibroblasts were cultured at 96-well flat-bottomed plates (TPP, 2 × 10^4^/well) in DMEM containing 10% fetal bovine serum and penicillin streptomycin 10^−4^ g/mL in a CO_2_ incubator at 37 °C, 95% humidity and 5% CO_2_ for 2 days [45].

### 4.3. Trypan Blue Assay

The counting of tumor cells in suspension and the determination of the cytotoxic effect of the chemotherapy drugs, NGF, PG-1, LL-37 and GAS were performed in a Goryaev chamber (MiniMed, Saint-Petersburg, Russia) under a light microscope (10× objective and 10× ocular, 100× total magnification).

To distinguish viable cells from dead cells, a 0.4% solution of trypan blue dye (Alfa Aesar, Beijing, China) at a 1:1 ratio was incubated at room temperature for 5 min. Then, 20 μL of the cell suspension was placed between the gap formed by the cover glass and the Goryaev chamber [46]. The cells containing the dye were considered dead; without the dye, they were considered alive. The concentration was determined as the number of cells in a unit volume of 1 mL = 1 cm^3^. In accordance with this, the calculation was carried out according to Equation (1):(1)$$X=\frac{B}{N\times V}\times Y=\frac{B}{N}\times 250000\times Y$$
where *X* is cell concentration per mL; *B*—the number of cells in *N* large squares of the chamber; *N* is the number of large squares to be counted; *V* is the volume of liquid over one large square (V = 1/250,000 mL); Y—dilution rate.

The cytotoxic antitumor effect of chemotherapy drugs, NGF, PG-1, LL-37 and GAS was assessed by the degree of the suppression of tumor cell growth, which was expressed by the cytotoxicity index (IC) calculated according to Equation (2):(2)$$N\left(\%\right)=\frac{1-sample}{control}\times 100$$
where *N*(%) is the IC of drugs, *sample* is the cell survival rate under the action of chemotherapy, NGF, PG-1, LL-37 and GAS and their combinations with chemotherapy, and *control* is the cell survival rate in the control sample [46].

### 4.4. MTT Assay

The cytotoxicity of the chemotherapeutic drugs NGF, PG-1, LL-37 and GAS in relation to C6 glioma cells was also assessed using the MTT assay [47]. For this purpose, 1 × 10^4^ cells/well, diluted in 50 μL of the DMEM medium, were added to the wells of sterile 96-well flat-bottomed plates 1 day before the treatment with chemotherapy drugs, peptides and streptococci. The concentration of cells in suspension was preliminarily determined as described above. Two-fold serial dilutions of GAS, NGF, PG-1, LL-37 and 2–10-fold dilutions of chemotherapy drugs in the DMEM medium were prepared, after which 50 μL of the resulting solutions were added to the wells of cell culture plates.

For each concentration of the reagent, three replicates were made. In the case of a study of the combined effect of two substances, 25 μL of each of them were added at the appropriate concentration. As a positive control, 50 μL of the DMEM medium was added to the wells with tumor cells instead of the reagent. As a negative control (0% viable cells), 100 μL of the DMEM medium was added to the empty wells of the plate, simulating the absence of living cells. The final sample volume in all wells was 100 μL.

Plates with samples were incubated for 1 day in a CO_2_ incubator at 37 °C and 5% CO_2_. After 1 day, 25 μL of MTT solution prepared with PBS at a concentration of 5 mg/mL was added to the wells and incubated for 3 h under the same conditions. At the end of incubation, 50 μL of isopropanol containing 0.04 M HCl was added to all samples and thoroughly mixed until the formazan precipitate was completely dissolved. The optical density of the solution in the wells of the plate was measured at a wavelength of 540 nm, subtracting the optical density at 690 nm, as background, using a SpectraMax 250 plate spectrophotometer (Molecular Devices, Sunnyvale, CA, USA). Experimental data were collected using SoftMax Pro 5.2 software (Molecular Devices, Sunnyvale, CA, USA).

On the basis of the collected data, the cytotoxicity of the active substances was determined. The percentage of dead cells was calculated based on the comparison of the optical density of samples with positive (100% viable cells) and negative (0% viable cells) controls according to Equation (3) [48]:
(3)$$DC(\%)=\frac{OD(control)-OD(sample)}{OD(control)-OD(0\%VC)}\times 100$$
where *DC*(%) is the percentage of dead cells in the sample, *OD(sample)* is the optical density of the sample containing the test substance in a given concentration; *OD*(0%*VC*) is an average optical density of wells with a culture medium that does not contain cells.

### 4.5. Determination of IC50 Dose, Combination Index and Combination Effects

To assess the cytotoxic antitumor efficacy of NGF, PG-1, LL-37, and chemotherapeutic agents on a C6 glioma cell culture, their dose to cause a 50% inhibition of cell viability (IC_50_) was calculated. For the MTT and trypan blue exclusion assays, the C6 cells were treated with PG-1 in concentrations of 2, 4, 8, 16, 32, and 64 μM, with LL-37 in concentrations of 0.5, 1, 2, 4, 8, and 16 μM, and with NGF in concentrations of 2.26 × 10^−4^, 3.7 × 10^−4^, 9.4 × 10^−4^, 1.88 × 10^−3^, 3.7 × 10^−3^, and 7.5 × 10^−3^ μM. We, therefore, used concentrations of NGF, LL-37 and PG-1 that led to 50% cell death (IC_50_) on rat glioma cell line C6. We compared the IC_50_s of NGF, LL-37, and PG-1 with the IC_50_ values of the chemotherapy drugs cisplatin, carboplatin, doxorubicin, temozolomide, and etoposide. Cisplatin was tested at concentrations of 1.66 × 10^−3^, 8.3 × 10^−4^, 3.32 × 10^−4^, 1.66 × 10^−4^, 8.3 × 10^−5^, 3.32 × 10^−5^; carboplatin was tested at concentrations of 2.69 × 10^−2^, 2.69 × 10^−3^, 1.35 × 10^−3^, 6.73 × 10^−4^, 2.69 × 10^−4^, 1.34 × 10^−4^; doxorubicin was tested at concentrations of 9.20 × 10^−4^, 4.60 × 10^−4^, 7.36 × 10^−5^, 3.68 × 10^−5^, 1.84 × 10^−5^, 7.36 × 10^−6^; temozolomide was tested at concentrations of 1.55 × 10^−2^, 5.15 × 10^−3^, 1.55 × 10^−3^, 7.73 × 10^−4^, 3.86 × 10^−4^, 1.55 × 10^−4^; and etoposide was tested at concentrations of 2.7 × 10^−5^, 1.35 × 10^−5^, 6.79 × 10^−6^, 3.39 × 10^−6^, 1.69 × 10^−6^, 8.49 × 10^−7^ M. Accordingly, we used NGF at 2.3 × 10^−4^ μM, LL-37 at 1 μM, and PG-1 at 10 μM concentrations in both the trypan blue exclusion and MTT assays, respectively. The IC_50_ values of NGF, PG-1, LL-37, chemotherapy drugs and their combinations were calculated by nonlinear regression analysis based on tested concentrations and corresponding effects using the OriginPro 8.5 software (OriginLab Corporation, Northampton, MA, USA).

The IC_50_ values of the chemotherapeutic drug in each combination was determined according to fixed proportions, for each pair of the chemotherapy drug according to Equation (4):(4)IC50(substance)=IC(combination)×W
where *W* is the proportion of a chemotherapy drug in the combination.

Having calculated the IC_50_ values of chemotherapy drugs, PG-1, LL-37, NGF and their combinations, we determined the combination index (CI) and the types of combination effects (synergism, antagonism, additivism) using Equation (5) [49].
(5)$$CI=\frac{D1}{\left(Dx\right)1}+\frac{D2}{\left(Dx\right)2}$$
where *CI* is the combination index, *D*1 and *D*2 are the IC50 doses of substances 1 and 2, causing cell death when used alone; (*Dx*)1 and *(Dx)*2 are the IC_50_ doses of these substances in combination. The IC_50_ of the substance in combination was calculated according to Equation (1).

The effect of a combination (*EC*) was considered to be additive if it was less than the sum of the effects of the combinants, but greater than the effect of the more active combinant (Equation (6)):(6)A+B>EC, but AB>Emax (A or B).
where *Emax* is the maximum effect of the substances when used alone.

The effect of *EC* was considered to be synergistic if it was less than the total effect of the combined substances of equal effect, but greater than the effects of one of the substances (Equation (7)):(7)A or B<EC, but AB<E∑(A+B)

A decrease in the effect (antagonism) was considered such if it was less than the effect of a more active compound (Equation (8)):(8)EC AB<Emax (A or B)

For combinations showing synergy, the dose reduction index (*DRI*) was also calculated using Equation (9):(9)$$DRI=\frac{D1}{\left(Dx\right)1}$$

### 4.6. Real-Time Cytotoxicity Analysis

For the real-time detection of the cytotoxic effects of GAS strains, NGF, PG-1, and LL-37 on C6 cell line, the xCELLigence system (Agilent, Santa Clara, CA, USA) was used. The RTCA iCELLigence instrument with E-Plate L8 were used for the experiments [50]. C6 cell line was prepared routinely using aseptic techniques. C6 cell lines were detached from plates using trypsin and 50,000 CU were plated on E-Plate L8 in 300 µL of DMEM. Loaded plates were cultivated in 5% of CO_2_ in the xCELLigence system during 24 h when 10^6^ CU of *Streptococcus pyogenes* strains, temozolomide, NGF, PG-1, and LL-37 (in IC50 doses) and their different combinations were added. Overnight cultures of *S. pyogenes* strains GUR, GURSA1, 7, and 21 were washed in PBS and then resuspended in DMEM to obtain the equal final concentration. A total of 100 µL of GAS strains resuspended in DMEM were placed on Todd-Hewitt agar plates and cultivated at 37 °C for 24 h to evaluate the cultures’ purity.

### 4.7. Reagents

Human cathelicidin LL-37 (AnaSpec, Fremont, CA, USA); porcine protegrin-1 (PG-1), (SynPep, Dublin, CA, USA); 7S of nerve growth factor from murine submaxillary gland (Sigma-Aldrich, Darmstadt, Germany); gentamicin sulfate (solution for intravenous and intramuscular administration, 40 mg/mL, Shandong Weifang Pharmaceutical Factory Co., Liaocheng, China); Doxorubicin-LANS^®^ (solution for intravascular and intravesical administration, 2 mg/mL, 5 mL, Veropharm, Moscow, Russia); Carboplatin-LANS^®^ (concentrate for the preparation of the solution for infusion, 10 mg/mL, 5 mL, Veropharm, Moscow, Russia); temozolomide (Temodal capsules, 100 mg, Orion Corporation, Orion Pharma, Espoo, Finland); Cisplatin-LANS^®^ (solution 0.5 mg/mL, 50 mL, Veropharm, Moscow, Russia); etoposide (Ebeve) (20 mg/mL, solution for infusion 10 mL, Ebewe Pharma, Unterach am Attersee, Austria).

### 4.8. Statistical Analysis

All experiments were performed at least in triplicate. The statistical significance of the differences between the means of different treatments and their respective control groups was determined using Student’s *t*-test. Data were counted with the standard deviation and considered significant at *p* < 0.05. To compare the differences between two independent groups with the small number of samples (*n* < 30), the nonparametric Mann–Whitney U-test was used [51]. The descriptive statistics were performed using the GraphPad Prism software (version 6.01, 09.21.2012, company, San Diego, CA, USA).

## 5. Conclusions

The live streptococcal strains *S. pyogenes* 21, *S. pyogenes* 7, *S. pyogenes* GUR, and *S. pyogenes* GURSA1, nerve growth factor (NGF) and the cathelicidin family antimicrobial peptides LL-37 and PG-1 show strong oncolytic effects, superior to the effect of chemotherapeutic drugs on C6 glioma cells in vitro.

Combinations of *S. pyogenes* 21 with NGF and *S. pyogenes* 21 with LL-37 have a pronounced cytostatic effect on C6 glioma cells, but less so than if administered separately.

For the combinations of PG-1 with carboplatin and LL-37 with etoposide, the synergism of their antitumor cytotoxic effect on C6 glioma cells was revealed according to the results of the MTT assay, whereas in the test with trypan blue, combinations of PG-1 with doxorubicin and cisplatin exhibited synergistic cytotoxic effects on the C6 glioma cell cultures.

PG-1, LL-37, NGF and *S. pyogenes* strains 7, 21, GUR, and GURSA1 do not have a statistically significant cytotoxic effect on cell cultures of normal human fibroblasts.

## Figures and Tables

**Figure 1 molecules-27-00569-f001:**
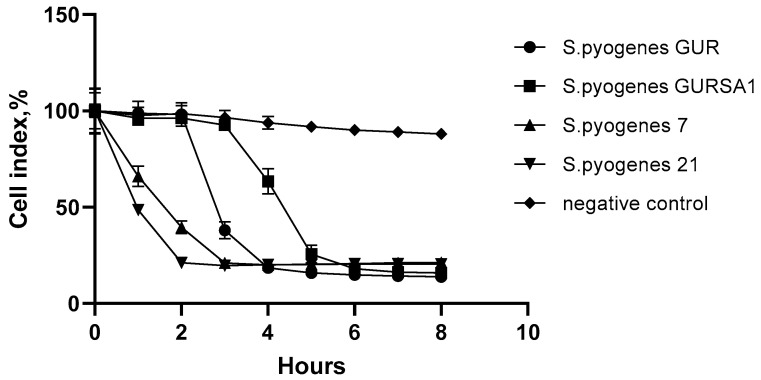
The effect of GAS strains on C6 cells in real time.

**Figure 2 molecules-27-00569-f002:**
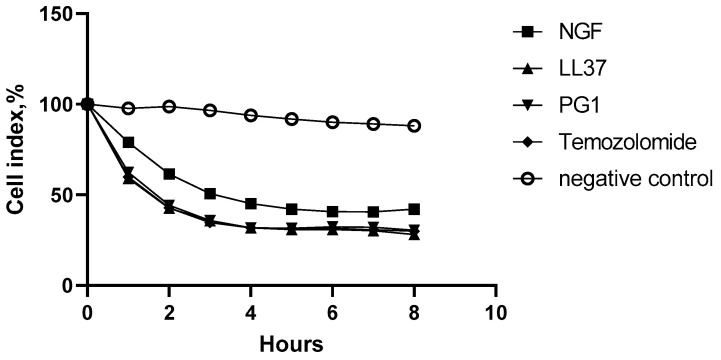
The effect of LL-37 (1 μM), PG-1 (10 μM), NGF (2.3 × 10^−4^ μM) and TMZ (1.55 μM) on C6 cells in real time.

**Figure 3 molecules-27-00569-f003:**
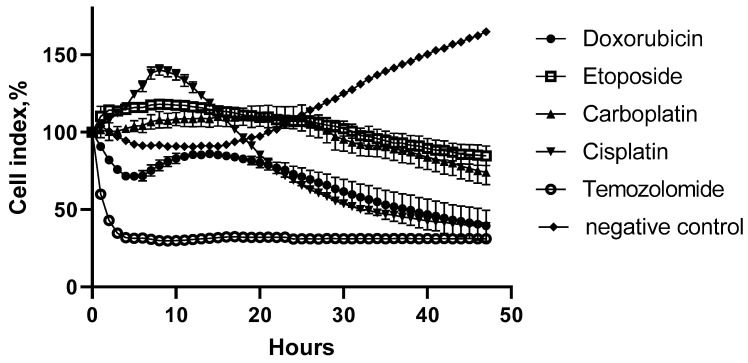
The effect of doxorubicin (50 μM), carboplatin (500 μM), cisplatin (125 μM), etoposide (2.5 μM) and temozolomide (1.55 μM) on C6 cells in real time.

**Figure 4 molecules-27-00569-f004:**
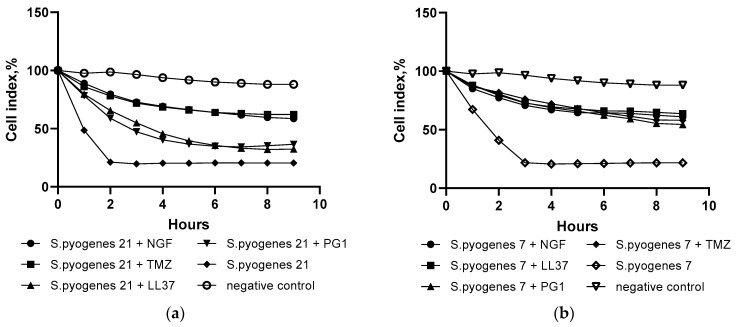
The effects *S. pyogenes* 7 (**a**) and *S. pyogenes* 21 (**b**) in combination with LL-37 (1 μM), PG-1 (10 μM), NGF (2.3 × 10^−4^ μM) and TMZ (1.55 mkM) on C6 cells in real time.

**Figure 5 molecules-27-00569-f005:**
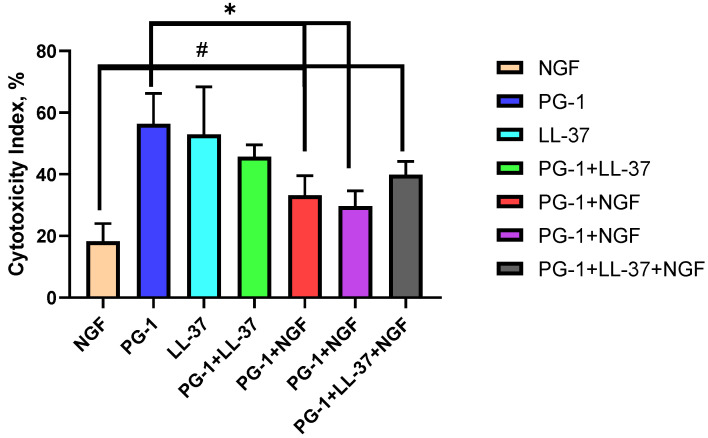
IC of the combinations of peptides LL-37 (1 μM), PG-1 (10 μM) with NGF (2.3 × 10^−4^ μM) on C6 glioma cells according to the results of the MTT assay; * *p* < 0.05 for the IC of the combinations and the IC of PG-1; # *p* < 0.05 for the IC of the combinations and the IC of NGF.

**Figure 6 molecules-27-00569-f006:**
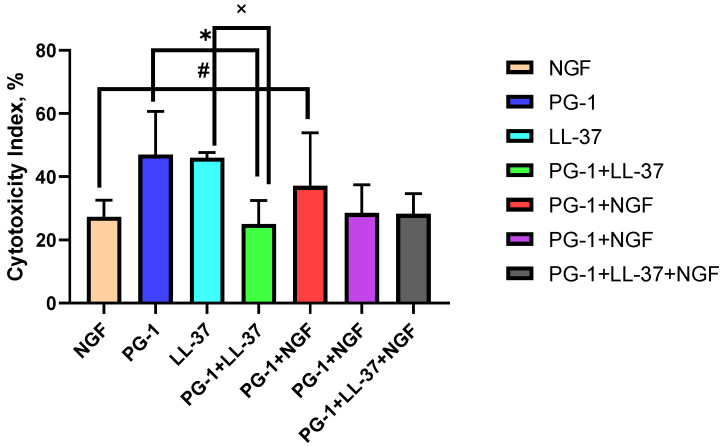
IC of the combinations of peptides LL-37 (1 μM) and PG-1 (10 μM) with NGF (2.3 × 10^−4^ μM) on C6 glioma cells according to the results of the trypan blue staining; * *p* < 0.05 for the IC of the combinations and the IC of PG-1; # *p* < 0.05 for the IC of the combinations and the IC of NGF; × *p* < 0.05 for the IC of combinations and the IC of LL-37.

**Figure 7 molecules-27-00569-f007:**
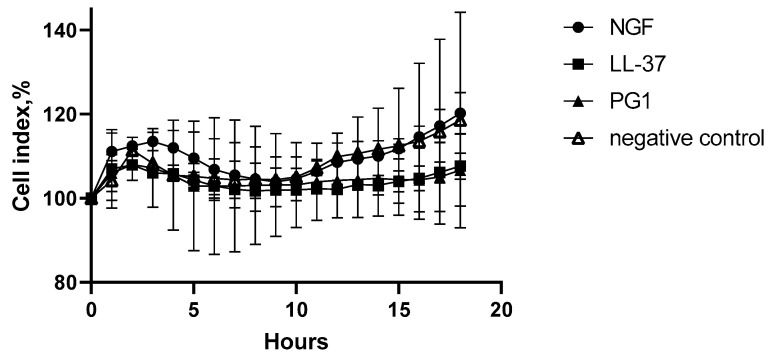
The effects of NGF (2.3 × 10^−4^ μM), LL-37 (1 μM), and PG-1 (10 μM) on human fibroblasts in real time.

**Figure 8 molecules-27-00569-f008:**
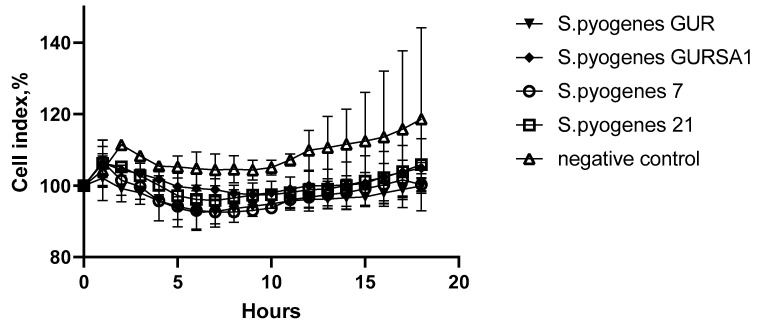
The effects of *S. pyogenes* 7, *S. pyogenes* 21, *S. pyogenes* GUR and *S. pyogenes* GURSA1 on human fibroblasts in real time.

**Table 1 molecules-27-00569-t001:** Cytotoxic effect of GAS on C6 glioma cells according to the MTT and the trypan blue test.

GAS Strain	Index of Cytotoxicity, %	
	MTT Assay	Trypan Blue Assay
*S. pyogenes* GUR	28.4 ± 0.07	52.6 ± 9.1
*S. pyogenes* GURSA1	19.2 ± 0.05	69.7± 10.5
*S. pyogenes* 7	32.2 ± 0.01	71.8± 11.9
*S. pyogenes* 21	79.1 ± 10.5 *	79.2 ± 14.5 *

* *p* < 0.05 between *S. pyogenes* 21 and *S. pyogenes* GUR. The death rate in the control group was 6.6%. Data are presented as mean ± standard deviation.

**Table 2 molecules-27-00569-t002:** IC50 of the one-day cytotoxic effect of chemotherapy drugs, NGF, LL-37, and PG-1 on C6 glioma cell cultures according to the MTT assay and trypan blue staining.

Substance	IC50, μM (MTT Assay)	IC50, μM (Trypan Blue Assay)
Doxorubicin	227.0 ± 9.9	30.0 ± 8.6
Carboplatin	244.1 ± 52.0	2600.3 ± 86.0
Temozolomide	391.5 ± 18.0	134.8 ± 63.7
Cisplatin	81.2 ± 12.1	82.2 ± 3.1
Etoposide	11.5 ± 2.5	1.8 ± 0.4
NGF	0.0148 ± 0.0019	0.0025 ± 0.001
LL-37	1.1 ± 0.1	1.1 ± 0.2
PG-1	10.1 ± 1.8	8.6 ± 3.6

Data are presented as mean ± standard deviation (*n* = 3).

**Table 3 molecules-27-00569-t003:** IC50 of the one-day cytotoxic effect of the chemotherapy drugs, NGF, LL-37, PG-1 and their combinations on C6 glioma cells according to MTT assay results.

Substance	Substances, μM	PG1 + Chemotherapy, μM	LL37 + Chemotherapy, μM	NGF + Chemotherapy, μM
Doxorubicin	227.0 ± 9.9	375.5 ± 14.6	172.6 ± 31.8	657.1 ± 26.1
Carboplatin	244.1 ± 52.0	1144.4 ± 464.1	55.2 ± 1.2	2847.8 ± 327.6
Temozolomide	391.5 ± 18.0	1313.4 ± 107.1	1805.1 ± 115.7	5855.2 ± 525.1
Cisplatin	81.2 ± 12.1	644.7 ± 16.85	4.0 ± 0.3	1167.0 ± 137.3
Etoposide	11.5 ± 2.5	27.1 ± 4.9	0.62 ± 0.14	36.4 ± 2.6
NGF	0.0148 ± 0.0019	-	-	-
LL-37	1.1 ± 0.1	-	-	-
PG-1	10.1 ± 1.8	-	-	-

Data are presented as mean ± standard deviation (*n* = 3).

**Table 4 molecules-27-00569-t004:** IC50 of the one-day cytotoxic effect of the chemotherapy drugs, NGF, LL-37, PG-1 and their combinations on C6 glioma cells according to the results of trypan blue staining.

Substance	Substances, μM	PG1 + Chemotherapy, μM	LL37 + Chemotherapy, μM	NGF + Chemotherapy, μM
Doxorubicin	30.0 ± 8.6	9.16 ± 2.2	1657.9 ± 124.9	701.1 ± 60.6
Carboplatin	2600.3 ± 86.0	2583.1 ± 106.2	4216.4 ± 171.7	23,296.4 ± 578.0
Temozolomide	134.8 ± 63.7	1300.4 ± 424.6	23,851.0 ± 238.7	23,296.4 ± 578.0
Cisplatin	82.2 ± 3.1	28.2 ± 12.29	1527.3 ± 168.2	4199.0 ± 737.3
Etoposide	1.8 ± 0.4	13.7 ± 0.7	27.0 ± 4.6	28.6 ± 2.2
NGF	0.0025 ± 0.001	-	-	-
LL-37	1.1 ± 0.2	-	-	-
PG-1	8.6 ± 3.6	-	-	-

Data are presented as mean ± standard deviation (*n* = 3).

**Table 5 molecules-27-00569-t005:** CI of the combined one-day exposure of chemotherapy drugs with NGF, LL-37, and PG-1 on C6 glioma cells according to MTT assay results.

Substance	PG1 + Chemotherapy	LL37 + Chemotherapy	NGF + Chemotherapy
Doxorubicin	2.72 (antagonism)	5.57 (antagonism)	2.90 (antagonism)
Carboplatin	0.11 (synergism)	6.90 (antagonism)	11.67 (strong antagonism)
Temozolomide	1.0 (addittivity)	13.3 (antagonism)	4.45 (antagonism)
Cisplatin	7.75 (antagonism)	1.53 (antagonism)	1413.8 (strong antagonism)
Etoposide	2.41 (antagonism)	0.57 (synergism)	4.44 (antagonism)

**Table 6 molecules-27-00569-t006:** CI of the combined one-day exposure of chemotherapy drugs with NGF, LL-37, and PG-1 on C6 glioma cells according to the results of trypan blue staining.

Substance	PG1 + Chemotherapy	LL37 + Chemotherapy	NGF + Chemotherapy
Doxorubicin	0.65 (synergism)	61.98 (strong antagonism)	24.69 (strong antagonism)
Carboplatin	4.49 (antagonism)	12.58 (strong antagonism)	5.78 (antagonism)
Temozolomide	9.20 (antagonism)	60.83 (strong antagonism)	62.11 (strong antagonism)
Cisplatin	0.60 (synergism)	13.97 (strong antagonism)	55.49 (strong antagonism)
Etoposide	4.43 (antagonism)	17.24 (strong antagonism)	18.07 (strong antagonism)

## Data Availability

The data presented in this study are openly available in Figshare at 10.6084/m9.figshare.16879432 (accessed on 18 September 2021).

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
