# Peer review of "In Vitro Evaluation of the Cytotoxic Effect of Streptococcus pyogenes Strains, Protegrin PG-1, Cathelicidin LL-37, Nerve Growth Factor and Chemotherapy on the C6 Glioma Cell Line"

_molecules, 2022, doi:10.3390/molecules27020569_

Round 1

Reviewer 1 Report

In this manuscript entitled “In vitro evaluation of the cytotoxic effect of Streptococcus pyogenes strains, Protegrin PG-1, Cathelicidin LL-37, Nerve Growth Factor and chemotherapy on C6 glioma cell line”, Suvorov and co-workers reported the use of Streptococcus pyogenes strains as effective anticancer agents towards C6 glioma cell lines. Moreover, the cytotoxicity studies were performed also in combination with PG-1, LL-37, NGF and some traditional and clinically-used chemotherapies.

Although progress made in the search for new and more effective anticancer therapies, the “weapons” to fight neurological tumors still remain very scarce. In this view, the authors presented here a very interesting work in which they propose the use of bacterial strains (and the combinations with other molecules) as promising anticancer agents. In my opinion, this is the real strength of this work.

The experimental part, though routinary, has been properly conceived and performed. The data have been clearly presented and the conclusions are consistent with the initial hypothesis.

In order to increase the overall readability and clarity of this paper, I would like to suggest to the authors to shorten the “Discussion” paragraph. In the present form, it appears as a mix of results discussion and a brief review of results by other authors. I think that this should be avoided if not strictly necessary and the discussion should be focused mainly on the paper’s results.

Some other minor issues:

  • All the footnotes in the tables should report the number of replicates and the data should have the standard deviation values.
  • The captions of figure 7 and 8 should be “The effects of NGF (2.3×10-4 μM), LL-37 (1 μM), PG-1 (10 μM) on normal human fibroblast cells in real time” and “The effects of S. pyogenes 7, S. pyogenes 21, S. pyogenes GUR and S. pyogenes GURSA1 on normal human fibroblast cells in real time”, respectively.
  • The authors, in particular the corresponding authors, should indicate an institutional email address, if possible.

Author Response

Dear Reviewer,

Thank you for your work on reviewing our manuscript, for the valuable comments and suggestions.

We shortened the “Discussion” as you suggested to focus it mostly on our results.

We added the information about the number of replicates in the tables as well as standard deviation values for the data.

We corrected captions of figures 7 and 8 and added an institutional email address.

All the corrections in the manuscript are made in edit mode using “Track Changes” function. 

Sincerely,

Authors  

Reviewer 2 Report

In the current study, authors have explored the use of bacterial strains of S. pyogenes, and antimicrobial peptides LL-37, PG-1 and NGF either individually or in combination with chemotherapeutic drugs for in-vitro oncolytic activity on the C6 glioblastoma cells. Cell death was measured either using MTT assay or trypan blue staining. MTT assay is a sensitive and well-established method to quantify cell death, and corroborated well with trypan blue. All strains used in the study had similar end cytotoxicity on C6 glioblastoma, and two strains of S.pyogenes, 21 and 7, were fast acting that started showing effect within the first hour of application. No synergy in cytotoxicity was observed with S.pyogenes in combination with both chemotherapeutics and peptides as one would expect. PG-1 and LL-37 showed synergistic effect when used in combination with carboplatin and etoposide respectively. The study adds to the growing literature on the use of bacterial strains and antimicrobial peptides as oncolytic activity. 

Specific comments:

One of the main hindrances for the use of chemotherapeutics, as rightly pointed by the authors, is their inability to cross blood-brain-barrier. Can authors comment on the ability of anti-microbial peptides and bacterial strains in crossing BBB? Also, can authors comment the clinical feasibility of using live bacterial strains in treating cancer and specifically glioblastoma?

There is significant English correction, with the use of colloquial English in some sections.  

Section 2.1: Authors note that S.pyogenes 21 has the highest cytotoxic effect on C6 glioblastoma cells. Can the authors specify the initial concentration of bacterial suspension used in this assay? since there is no mention of initial count other than 100ul of the overnight culture. Although this could be the real difference, the difference could also arise from different starting inoculum of the bacterial strains.

What is the significance of 2.5hr for bacterial inhibition? When it appears that all the bacterial strains have similar oncolytic activity after 6 hr, why is it that strains 21 and 7 is considered more effective than the rest of the strains.

Section 2.2: can authors specify at what concentrations of these peptides are toxic to humans in clinical settings? Although this study is done in an in-vitro setting, it would give a perspective on the concentrations used.

Section 2.4: Is this not expected? Did the authors measure the viability of the microbial strains after the experiment, since the peptides are anti-microbial? Again, what is the significance of fast cytotoxicity versus end cytotoxicity?

Section 2.5: Can the authors explain the discrepancy between MTT result and trypan blue? MTT assay showed a synergy for PG-1 with carboplatin and LL-37 with etoposide, while using trypan blue PG-1 showed synergy with doxorubicin.

Section 2.6: The cytotoxicity seems to be specific to the glioblastoma cell line and not to normal fibroblasts. Correct legends of Fig. 7 and Fig. 8 from use of C6 cells to fibroblast cells.

Author Response

Dear Reviewer,

Thank you for your work on reviewing our manuscript, for the interesting questions and valuable comments.

Question 1.

One of the main hindrances for the use of chemotherapeutics, as rightly pointed by the authors, is their inability to cross blood-brain-barrier (BBB). Can authors comment on the ability of anti-microbial peptides and bacterial strains in crossing BBB? Also, can authors comment the clinical feasibility of using live bacterial strains in treating cancer and specifically glioblastoma?

The size of anti-microbial peptides are 18 and 37 amino acids and in healthy person they can’t cross BBB as it is possible for <500 Da peptides. But it has been shown that patients with brain tumors (as well as patients with neurodegenerative disorders) have violation of the BBB which facilitates the penetration for larger substances. The development of brain tumors usually is accompanied by the inflammation in the nervous tissue. In brain tumors, as in meningitis, BBB penetrance increases due to the secretion of proinflammatory cytokines, inhibition of the expression of adhesion molecules on endothelial cells and dismemberment of slit contacts with the formation of fenestrations.

As streptococci are among few microorganisms able to cause meningitis, we suppose that the cells are able to cross BBB. One of our scientific goal is to achieve gene-modified avirulent oncolytic streptococcal strain to use as antitumor agent.

The use of live bacteria for cancer treatment has long history since 1891 when William B. Coley started to use S. pyogenes for sarcoma treatment. We believe animal model study will give us more detailed information how to apply streptococci (alive bacteria as well as active bacterial compounds) to achieve effective glioblastoma treatment.          

Question 2.

There is significant English correction, with the use of colloquial English in some sections.

We will contact MDPI for English language editing service

Question 3.

Section 2.1: Authors note that S.pyogenes 21 has the highest cytotoxic effect on C6 glioblastoma cells. Can the authors specify the initial concentration of bacterial suspension used in this assay? since there is no mention of initial count other than 100ul of the overnight culture. Although this could be the real difference, the difference could also arise from different starting inoculum of the bacterial strains.

For all the experiments 106 CU of cells washed from media and resuspended in DMEM were used. Information is added to 4.1 section of material and methods.

Question 4.

What is the significance of 2.5hr for bacterial inhibition? When it appears that all the bacterial strains have similar oncolytic activity after 6 hr, why is it that strains 21 and 7 is considered more effective than the rest of the strains.

Difference in the speed of cytotoxic activity of the strains points on different mechanisms of they oncolytic action. Apparently, S.pyogenes 21 and S.pyogenes 7 are able to produce extracellular oncolytic substances which are absent in S.pyogenes Gur and GURSA1 (for example erythrogenic toxin A). As these extracellular substances depleted all the strains achieve similar end cytotoxicity. In other words – 7 and 21 have some advantage which is expressed in their faster oncolytic action that is why we characterize S.pyogenes 21 and 7 as having highest cytotoxic effect on C6 cells.     

Question 5.

Section 2.2: can authors specify at what concentrations of these peptides are toxic to humans in clinical settings? Although this study is done in an in-vitro setting, it would give a perspective on the concentrations used.

According to the literature the peptides are considered toxic at 100 µM and higher.

Question 6.

Section 2.4: Is this not expected? Did the authors measure the viability of the microbial strains after the experiment, since the peptides are anti-microbial? Again, what is the significance of fast cytotoxicity versus end cytotoxicity?

Yes, we expected that peptides would affect bacterial cells. As we noticed in preliminary studies bacterial lysates can also be oncolytic but less than live bacteria. So, we were interested in the effect of the combination of the peptides and bacteria even knowing that streptococci will be dead. We didn’t measure the viability of the microbial strains after the experiment.

Fast cytotoxicity indicates that the strain has some active extracellular structures worth detailed studies.

Question 7.

Section 2.5: Can the authors explain the discrepancy between MTT result and trypan blue? MTT assay showed a synergy for PG-1 with carboplatin and LL-37 with etoposide, while using trypan blue PG-1 showed synergy with doxorubicin.

The differences between the IC50 values in MTT assay and the test with trypan blue for the same combinations can reflect the different degree of involvement of cellular processes with mitochondrial dehydrogenases as well as damage to the cell membrane, assessed in these tests. In case the IC50 of the combination in the MTT assay was lower than that for staining with trypan blue, such as PG-1 or LL-37 with doxorubicin, it indicates the role of mitochondrial dehydrogenases in the realization of the cytotoxic effects of these combinations of substances. On the contrary, when the IC50 of the combination in the MTT test was higher than the IC50 in the test with trypan blue, for example, PG-1 with cisplatin, or etoposide, this indicates the predominant participation of cell plasmalemma damage in the cytotoxic action of the combination.

Question 8.

Section 2.6: The cytotoxicity seems to be specific to the glioblastoma cell line and not to normal fibroblasts. Correct legends of Fig. 7 and Fig. 8 from use of C6 cells to fibroblast cells.

Corrected

All the corrections in the manuscript are made in edit mode using “Track Changes” function.   

Sincerely,

Authors  

This manuscript is a resubmission of an earlier submission. The following is a list of the peer review reports and author responses from that submission.

Round 1

Reviewer 1 Report

Dear Authors,

Your manuscript on the combinatory treatment of C6 glioma cells with new treatment options is an interesting approach for alternative glioma therapies that are urgently needed. You nicely describe the combinatory effects and give reasonable explanations for the results obtained. Nevertheless I do see some points which could be improved.

  1. The introduction should be worked over a bit. In line 47-50 the sentences sound like they were just placed one after another. Thereafter (line 52) you mention bevacizumab to be a chemotherapeutic, which is definitely not the case as it is a therapeutic antibody against VEGF. Also the description of the mode of action of cisplatin and temozolomide is not correct. Both share the same target in the DNA but don’t work as described.
  2. It is not clear to me which concentrations of drugs where used in which experiments. The figure legends should include this information. Also the treatment curves of MTT and trypan blue measurements should be made available as supplementary data.
  3. Is there any particular reason to use C6 glioma cells? In the current form of the study it would make more sense to use human cells, to be closer to the intended clinical application. To me it would make sense to use rat cells if also normal rat cells, neural stem cells or astrocytes, would be used as healthy control. By doing so you would make the whole study a lot more interesting as it would also give information on possible strength of side effects.

Best regards

Reviewer 2 Report

This manuscript describes cytotoxic effects of Streptococcus pyogenes strains, Protegrin 2 PG-1, Cathelicidin LL-37, NGF and their combinations with chemotherapy on the glioma cell line C6. While in general the topic of combining different therapeutic principle to enhance efficacy is of great relevance in oncology, the approach take here is not sufficient to address this question appropriately. My main concerns are:

*The study uses one permanent cell line only. It is now firmly established that cell lines grown on plastic for years and decades represent very limited tumor models. Instead, the authors should use more authentic and more native biomaterial such as primary cultures of tumor cells from patients samples or organoids of these. If not available, at least three different cell lines should be used in the study to demonstrate that the effects seen are not limited to just this one cell line.

*Next to efficacy, toxicity is a major issue in oncology. Many treatments are limited regarding dosage due to substance toxicity. This study lacks any information on the effect of the drugs used on normal cells. The panel of cells used to test the substances and combinations thereof need to include at least one type of normal cells, such as fibroblasts, endothelial or other cells. Most of them are commercially available, some may easily be prepared from clinical material.

*The effects lack confirmation in an animal model.

*The journal supports an open data policy and asks all authors of articles to share their original, unprocessed research data in machine-readable form (e.g. csv or xlsx) in a public data repository such as Zenodo or FigShare. Please include a link to the data deposit at the end of the material and methods section.

*The method for the cell incubations with substances/bacteria is not provided. The method description include the origin of the substances used and their purity. How were bacteria cultured, how was their identity and purity controlled, how were they prepared for incubation with the cells, were they separated from supernatant/media before addition to the cells?

*For experiments described in Fig. 1-3 it is unclear, how many independent experiments are represented by the data. As no error bars are included, it seems they were carried out only once. To be representative and robust, experiments need to be performed at least 3 times independently. The number of replicates used in the individual experiment should be stated as well as the number of independent experiments. SD or SEM should be reported as error bars in the graphs. SD/SEM and mean calculation should be done as means of the mean: first calculate the mean of replicate values in the individual experiments, then used these means to calculate the aggregated mean and SD/SEM for the whole set of experiments. These should then be reported. Wherever meaningful, a statistical test should be performed, described and the results reported.

Author Response

Dear Reviewer,

Thank you for your detailed revision, valuable comments and suggestions. Your opinion has made us rethink some experiments design we already started to perform as extension to current work.

Further we would like to provide our answers to your comments:

*The study uses one permanent cell line only. It is now firmly established that cell lines grown on plastic for years and decades represent very limited tumor models. Instead, the authors should use more authentic and more native biomaterial such as primary cultures of tumor cells from patients samples or organoids of these. If not available, at least three different cell lines should be used in the study to demonstrate that the effects seen are not limited to just this one cell line.

We agree that the use of different cell lines will provide more information about the potential of the substances antitumor effect. Moreover, after we had got positive results with C6 cells we started to perform a new study with U251 glioma cells, A549 lung carcinoma and myelogenous leukemia K562 cell lines, also we collected a few samples of patients’ glioblastoma to continue the study. But even if we don’t get similar effects with other cell lines, we suggest that will not cancel results we have got with C6 cells. So, we suppose that the effect we demonstrated on C6 cell line worth a publication.

*Next to efficacy, toxicity is a major issue in oncology. Many treatments are limited regarding dosage due to substance toxicity. This study lacks any information on the effect of the drugs used on normal cells. The panel of cells used to test the substances and combinations thereof need to include at least one type of normal cells, such as fibroblasts, endothelial or other cells. Most of them are commercially available, some may easily be prepared from clinical material.

NGF, LL-37, PG-1 are endogenous substances which are synthesized and secreted by normal human and animal cells. Their effects on normal cells are well known.

NGF is a factor that supports neuronal survival:

Johnston M V, Rutkowski J L, Wainer B H, Long J B, Mobley W C. NGF effects on developing forebrain cholinergic neurons are regionally specific. Neurochem Res. 1987 Nov;12(11):985-94. doi: 10.1007/BF00970927.

Hatanaka H, Tsukui H, Nihonmatsu I. Developmental change in the nerve growth factor action from induction of choline acetyltransferase to promotion of cell survival in cultured basal forebrain cholinergic neurons from postnatal rats. Brain Res. 1988 Mar 1;467(1):85-95. doi: 10.1016/0165-3806(88)90069-7.

LL-37 and PG-1 action on normal cells have also been studied:

Yang, Y., Choi, H., Seon, M. et al. LL-37 stimulates the functions of adipose-derived stromal/stem cells via early growth response 1 and the MAPK pathway. Stem Cell Res Ther 7, 58 (2016). https://doi.org/10.1186/s13287-016-0313-4

Zharkova M.S., Orlov D.S., Golubeva O.Y., Chakchir O.B., Eliseev I.E., Grinchuk T.M., Shamova O.V. Application of Antimicrobial Peptides of the Innate Immune System in Combination With Conventional Antibiotics-A Novel Way to Combat Antibiotic Resistance? Front Cell Infect. Microbiol. 2019. Vol. 9: 128. doi: 10.3389/fcimb.2019.00128.

Also, NGF, LL-37, PG-1 action has been studied on other types of tumors:

Chen X., Zou X., Qi G., Tang Y., Guo Y., Si J., Liang L. Roles and Mechanisms of Human Cathelicidin LL-37 in Cancer. Cell Physiol. Biochem., 2018, Vol. 47, no. 3, pp. 1060-1073.   -doi: 10.1159/000490183.

Fan R., Tong A., Li X., Gao X., Mei L., Zhou L., Zhang X., You C., Guo G. Enhanced antitumor effects by docetaxel/LL37-loaded thermosensitive hydrogel nanoparticles in peritoneal carcinomatosis of colorectal cancer. Intern. J. Nanomedicine, 2015, Vol. 10, pp. 7291–7305.

Therefore, we decided not to repeat these studies, but focused on the effects of the substances towards brain tumor cells as these effects had never been studied before.

As to streptococcal effect on normal cells - some of it had been shown in our previous work:

Suvorova, M.A.; Kramskaya, T.A.; Duplik, N.V.; Chereshnev, V.A.; Grabovskaya, K.B.; Ermolenko, E.I.; Suvorov, A.N.; Kiseleva, E.P. The effect of inactivation of the M-protein gene on the antitumor properties of live Streptococcus pyogenes in experiment. Voprosi oncologii 2017, 63(5), 803-807.

Suvorova, M.A.; Tsapieva, A.N.; Kiseleva, E.P.; Suvorov, A.N.; Bak, E.G.; Arumugam, M.; Chereshnev V.A. Complete genome sequences of emm111 type Streptococcus pyogenes strain GUR, with antitumor activity, and its derivative strain GURSA1 with an inactivated emm gene. Genome Announcements 2017, 5(38), e00939-17. [https://pubmed.ncbi.nlm.nih.gov/28935732/]

Also, we have to mention, that live pathogenic streptococci will not be used as antitumor agent anytime for clinical application. But their avirulent derivates or recombinant bacterial peptides are of great potential. These substances which we are planning to get soon will sure be tested both in vitro and in vivo for adverse effects and toxicity.

*The effects lack confirmation in an animal model.

In vivo experiments are going to be the next step in our work. Work with animals requires ethical committee approval, new experiment design along with the animals itself as well as additional funding. For this year we have been focused on in vitro experiments. 

*The journal supports an open data policy and asks all authors of articles to share their original, unprocessed research data in machine-readable form (e.g. csv or xlsx) in a public data repository such as Zenodo or FigShare. Please include a link to the data deposit at the end of the material and methods section.

Data is downloaded and the link is provided.

*The method for the cell incubations with substances/bacteria is not provided. The method description include the origin of the substances used and their purity. How were bacteria cultured, how was their identity and purity controlled, how were they prepared for incubation with the cells, were they separated from supernatant/media before addition to the cells?

Information is added to Streptococcus pyogenes strains and Real-time cytotoxicity analysis sections at Materials and methods:

Streptococcus pyogenes strains. For the study 4 strains of GAS with previously discovered oncolytic properties were chosen - S. pyogenes strains GUR, GURSA1, 7, 21. S. pyogenes GUR is a throat isolate from a scarlet fever patient, which was kindly provided by prof. Chereshnev V.A., Perm State University. S. pyogenes GURSA1 is derivative S. pyogenes GUR strain with an inactivated M-protein encoding gene [25]. S. pyogenes strains GUR, GURSA1 previously shown cytotoxic activity towards murine malignant tumor cells [25,26]. S. pyogenes strains 7, 21 were selected during cytotoxic activity preliminary studies of several GAS strains towards C6, U-251 and А-549 cell lines. The strains were cultivated in Todd-Hewitt broth (Condalab, Spain) for 16 h at 37 0C from the single colony.    

Real-time cytotoxicity analysis. For real-time detection of cytotoxic effects of GAS strains, NGF, PG-1, and LL-37 on C6 cell line Agilent xCELLigence System was used. The RTCA iCELLigence instrument with E-Plate L8 were used for the experiments [57]. C6 cell line was prepared routinely using aseptic technique. C6 cell line were washed from plates using trypsin and 50000 CU were plated on E-Plate L8 in 300 µL of DMEM. Loaded plates were cultivated in 5% of CO2 in xCELLigence System during 24 h when 106 CU of Streptococcus pyogenes strains, temozolomide, NGF, PG-1, and LL-37 (in IC50 doses) and their different combinations were added. Overnight cultures of S. pyogenes strains GUR, GURSA1, 7, 21 were centrifuged at 6000 g, the pellets were separated from the media and were washed in PBS and than resuspended in DMEM to obtain equal final concentration. 100 µL of GAS strains resuspended in DMEM were placed on Todd-Hewitt agar (Condalab, Spain) plates and cultivated at 37 0C for 24 h to evaluate the cultures purity. 

*For experiments described in Fig. 1-3 it is unclear, how many independent experiments are represented by the data. As no error bars are included, it seems they were carried out only once. To be representative and robust, experiments need to be performed at least 3 times independently. The number of replicates used in the individual experiment should be stated as well as the number of independent experiments. SD or SEM should be reported as error bars in the graphs. SD/SEM and mean calculation should be done as means of the mean: first calculate the mean of replicate values in the individual experiments, then used these means to calculate the aggregated mean and SD/SEM for the whole set of experiments. These should then be reported. Wherever meaningful, a statistical test should be performed, described and the results reported.

In the manuscript we demonstrated the cytotoxic effect of peptides and GAS strains using three different methods. At first we performed MTT and trypan blue assays each 3 to 5 times. In the end to illustrate substances cytotoxicity during time we performed xCelligence experiments which gave us real-time curves of cell death dynamics. Obtained results proved previous data and we finished this set of experiments as for each substance we got results not only of three independent experiments but of three independent methods. For each experiment we used the same amount of C6 cells and GAS/peptides so all the data is comparable and reliable. Fig.1 we replaced with one containing error bars as this experiment was performed in replicates.  

We do hope you find our answers satisfactory and thank you again for your work.

Sincerely, A.

Reviewer 3 Report

Comments are in the uploaded text.

Author Response

Dear Reviewer,

Thank you for your work on reviewing our manuscript, for your comments and suggestions. We’ve made all the changes according to your report, please find them in the revised manuscript.

Sincerely, A.

Round 2

Reviewer 2 Report

As the majority of comments and concerns mentioned earlier remains without substantial improvement, I'm unable to recommend publication of this manuscript.